# B Cell Activation and Escape of Tolerance Checkpoints: Recent Insights from Studying Autoreactive B Cells

**DOI:** 10.3390/cells10051190

**Published:** 2021-05-13

**Authors:** Carlo G. Bonasia, Wayel H. Abdulahad, Abraham Rutgers, Peter Heeringa, Nicolaas A. Bos

**Affiliations:** 1Department of Rheumatology and Clinical Immunology, University Medical Center Groningen, University of Groningen, 9713 Groningen, GZ, The Netherlands; c.g.bonasia@umcg.nl (C.G.B.); w.abdulahad@umcg.nl (W.H.A.); a.rutgers@umcg.nl (A.R.); 2Department of Pathology and Medical Biology, University Medical Center Groningen, University of Groningen, 9713 Groningen, GZ, The Netherlands; p.heeringa@umcg.nl

**Keywords:** autoimmune diseases, B cells, autoreactive B cells, tolerance

## Abstract

Autoreactive B cells are key drivers of pathogenic processes in autoimmune diseases by the production of autoantibodies, secretion of cytokines, and presentation of autoantigens to T cells. However, the mechanisms that underlie the development of autoreactive B cells are not well understood. Here, we review recent studies leveraging novel techniques to identify and characterize (auto)antigen-specific B cells. The insights gained from such studies pertaining to the mechanisms involved in the escape of tolerance checkpoints and the activation of autoreactive B cells are discussed. In addition, we briefly highlight potential therapeutic strategies to target and eliminate autoreactive B cells in autoimmune diseases.

## 1. Introduction

Autoimmune diseases are mostly chronic, complex, immune disorders that vary in severity from mild to lethal. It has been estimated that up to 9.4% of the world population is affected by an autoimmune disease [1]. In addition, the prevalence of autoimmune diseases is increasing, making autoimmune diseases a major disease burden globally [2]. Multiple factors are thought to be involved in the development of autoimmune diseases including genetic and environmental factors [3]. Autoimmune diseases are characterized by a loss of self-tolerance leading to an immune response against self-antigens. The autoimmune response can cause inflammation and damage to specific or multiple tissues and organs, depending on the target autoantigen.

B cells play a key role in the pathogenesis of many autoimmune diseases (Table 1). The presence of autoantibodies, and their described contribution to the pathogenesis of these B cell-mediated autoimmune diseases, is proof of B cell dependency. However, the most conclusive evidence that B cells act as key players in B cell-mediated autoimmune diseases is derived from the successful treatment of patients with therapies that specifically deplete B cells, mainly rituximab, a chimeric anti-CD20 monoclonal antibody that targets CD20^+^ B cells [4]. In addition to antibody production by terminally differentiated B cells (plasma cells), B cells can also contribute to disease development and progression by antibody-independent mechanisms. B cells are capable of producing cytokines which can affect the T cell response [5,6]. Moreover, B cells act as professional antigen-presenting cells (APCs) [6].

Since B cells play a major role in autoimmune diseases, it is of high interest to understand the development of autoreactive B cells. However, our knowledge regarding autoreactive B cell development is rudimentary. During B cell development, immature B cells are generated in the bone marrow that express unique randomly assembled B cell receptors (BCRs). B cells mature in the periphery and B cell activation is initiated upon antigen binding. Activated B cells differentiate into memory and plasma cells, which provide protective immunity. As a consequence of the random process of assembling BCRs, a large proportion of the immature B cells generated in the bone marrow is autoreactive. To prevent autoimmunity, B cells are subjected to various self-tolerance checkpoints from the immature B cell stage until the plasma cell stage. Central self-tolerance checkpoint mechanisms in the bone marrow, in conjunction with self-tolerance checkpoint mechanisms in the periphery, normally prevent the occurrence of pathogenic autoreactive B cells. In autoimmune diseases, these self-tolerance checkpoints are breached; however, which exact checkpoints fail, and how this occurs, is still elusive. Moreover, the precursor cells from which pathogenic autoreactive B cells derive remain unknown. Furthermore, how autoreactive B cells are activated in disease is incompletely understood.

Research into autoreactive B cell development in humans has been hampered because techniques to detect, isolate, and characterize these low-frequency cells were lacking. In recent years, novel sophisticated and sensitive techniques have been developed that facilitate the study of low-frequency (auto)antigen-specific B cells [25].

In this review, we summarize our current knowledge pertaining to the development of autoreactive B cells with a main focus on the autoimmune diseases rheumatoid arthritis (RA), systemic lupus erythematosus (SLE), granulomatosis with polyangiitis (GPA), and pemphigus (Table 1). Following a brief overview of general B cell development and B cell tolerance, we discuss recent insights gained from studies focusing on the autoreactive B cells themselves and highlight their relevance for our understanding of disease pathogenesis and the design of novel therapeutic approaches.

## 2. General B Cell Development and B Cell Tolerance

### 2.1. B Cell Generation and Central Tolerance

B cell generation starts in the bone marrow, in which hematopoietic stem cells transit into immature B cells through a series of developmental steps. Immature B cells are the first cells to express a complete BCR during B cell development. Moreover, immature B cells express unique BCRs, and therefore each cell has a unique antigen specificity, as a result of the random recombination of variable (V), diversity (D), and joining (J) immunoglobulin gene segments [26]. Since the process of V(D)J recombination is random, a portion of immature B cells is autoreactive. In fact, it has been estimated that 75% of the immature B cells express BCRs that bind self-antigens [27]. Autoreactive immature B cells are subjected to central tolerance mechanisms in order to eliminate these cells from the B cell pool (Figure 1). These mechanisms include: (i) clonal deletion, i.e., removal of autoreactive B cells by the induction of apoptosis; (ii) anergy, i.e., a state of B cells characterized by unresponsiveness to antigens, downregulation of BCR expression, and a short life-span; and (iii) receptor editing, i.e., a mechanism that replaces the light chain of the BCR with a newly recombined light chain, resulting in a BCR with a different antigen specificity [28,29,30,31]. The fate of immature B cells is controlled by BCR signaling. Immature B cells with functionally unligated BCRs exhibit tonic BCR signaling, a constitutive BCR signal that is important for B cell survival and development. Moreover, whether immature B cells are subjected to central tolerance mechanisms is dependent on the strength of the interaction between the BCR and self-antigens. A moderate avidity BCR–self-antigen interaction stimulates cell survival and the continuation of development. Contrarily, a high avidity BCR–self-antigen interaction causes BCR signaling above tonic levels, resulting in developmental arrest and the subjection of cells to central tolerance mechanisms [28]. Of the total immature B cell population, around 40% are still autoreactive after the central tolerance checkpoint [27]. This relatively large remaining proportion of autoreactive immature B cells is probably because immature B cells solely encounter the self-antigens present in the microenvironment of the bone marrow. Furthermore, most autoreactivity at this stage is considered to be due to the presence of B cells that express BCRs with a low affinity for self-antigens.

### 2.2. Peripheral B Cell Maturation and Tolerance

Immature B cells emerge from the bone marrow into the periphery as transitional B cells which home to the spleen. In the spleen, transitional B cells acquire IgD expression in addition to IgM and develop into mature naive B cells. As part of peripheral tolerance, transitional B cells undergo clonal deletion or attain an anergic state in response to high avidity BCR–self-antigen interactions (Figure 2A) [6]. Since other self-antigens are expressed in the spleen compared to the bone marrow, the frequency of autoreactive B cells further reduces.

Naive B cells continuously circulate through the blood and lymphatic system and primarily populate follicular sites within lymph nodes and the spleen. Naive B cell activation is initiated upon antigen binding. Naive B cells take up the antigen and present peptides from the antigen via MHC-II to CD4^+^ T helper (T_H_) cells with the same antigen specificity. Additionally, naive B cells express B7 upon antigen binding. B7 binds to CD28 of MHC-II-peptide-bound antigen-specific T_H_ cells, leading to their activation. Subsequently, B7–CD28 interaction induces the expression of CD40L in T_H_ cells. Complete activation of naive B cells is achieved by CD40L costimulation from activated antigen-specific T_H_ cells [32,33]. In addition, other signals can aid in overcoming the BCR signaling threshold for complete naive B cell activation. These include the activation of Toll-like receptors (TLRs) and complement receptors (CRs) on naive B cells [34,35]. Concerning the elimination of autoreactive naive B cells, clonal deletion or anergy is induced in antigen-activated naive B cells when costimulation by T_H_ cells is absent (Figure 2A). Costimulation is absent as a result of the elimination of autoantigen-specific T cells by central tolerance mechanisms in the thymus and peripheral tolerance mechanisms. Ultimately, the autoreactive B cell pool is greatly reduced. It has been demonstrated that approximately 20% of the peripheral naive B cells are autoreactive [27]. However, this presumably represents a mix of low-affinity autoreactive naive B cells together with autoreactive naive B cells undergoing anergy or clonal deletion.

When naive B cells are fully activated, they proliferate and differentiate into short-lived plasma cells at extrafollicular sites within the lymph node or the spleen. Short-lived plasma cells predominantly express low-affinity IgM antibodies. Besides migration and development at extrafollicular sites, activated naive B cells migrate into germinal centers. There, the B cells proliferate and undergo somatic hypermutation (SHM) of their immunoglobulin V genes, which affects BCR affinity. B cells that have the highest affinity for antigen retained on follicular dendritic cells (FDCs), and that present antigen to CD4^+^ follicular T helper (T_FH_) cells with the same antigen specificity, are positively selected. These positively selected B cells receive costimulatory signals (CD40L) from T_FH_ cells to undergo repetitive rounds of proliferation, SHM, and selection [36,37]. Ultimately, positively selected high-affinity B cells undergo class switching and will mature into memory B cells and long-lived plasma cells [37,38]. Regarding autoreactivity and tolerance, SHM may lead to the generation of autoreactive B cells because this is a random process. The self-tolerance checkpoints after B cell activation are not well-defined. It has been described that B cells that do not receive costimulatory signals from T_FH_ cells are removed by clonal deletion (Figure 2B) [39]. The absence of costimulatory signals is the result of the purge of autoantigen-specific T_FH_ cells by T cell tolerance mechanisms during T cell development. Furthermore, as a byproduct of SHM, SHM-mediated receptor editing of autoreactive B cells can decrease the affinity for autoantigens, thereby removing autoreactivity [39].

While the majority of autoreactive B cells are deleted during B cell maturation, a small fraction of the normal B cell repertoire in healthy individuals is autoreactive. These non-pathogenic natural autoreactive B cells, which do not enter germinal centers, predominantly produce low-affinity polyreactive IgM antibodies and to a lesser extent low-affinity polyreactive IgG and IgA antibodies [40,41]. Various functions have been ascribed to natural autoreactive B cells, including clearance of cellular debris from apoptotic cells and inhibition of autoimmune inflammatory processes by masking self-antigens [40,41].

Overall, several self-tolerance checkpoints are present during normal B cell development, at various (micro)anatomical sites, and during different developmental stages, to prevent the occurrence of pathogenic autoreactive B cells.

## 3. Autoreactive B Cell Development in Autoimmune Diseases

### 3.1. Compromised Self-Tolerance Checkpoints and Precursor Cells of Autoreactive B Cells

Two issues in the field of autoimmunity that have remained unresolved thus far are the identification of the precise self-tolerance checkpoints that are breached and the characterization of the precursors from which autoreactive B cells derive. Recently, multiple studies have focused on the B cell subset distribution and phenotypical characterization of autoreactive B cells in autoimmune diseases and health, providing novel insights into the mechanisms underlying the development of autoreactive B cells.

Within the circulating pool of autoreactive B cells, an enrichment of memory/class-switched memory B cells has been demonstrated in RA, GPA, and pemphigus vulgaris patients when compared to the total B cell compartment of patients and/or the (natural) autoreactive B cell compartment of healthy individuals [42,43,44]. These observations imply a breach of self-tolerance within germinal centers in these diseases. Normally, a self-tolerance checkpoint exists within germinal centers in which autoreactive B cells are unable to develop in germinal centers due to the absence of T_FH_ cell support (Figure 2B). Hence, this would also suggest that the self-tolerance of T_FH_ cells is defective, or that other mechanisms are present that circumvent T_FH_ dependency. Moreover, these observations indicate that autoreactive B cells in RA, GPA, and pemphigus vulgaris derive from germinal centers since only memory/class-switched memory B cells were enriched and memory B cell generation and class switching mostly take place at these sites. Furthermore, autoreactive B cells may be generated from non-autoreactive B cells or non-pathogenic autoreactive B cells with a very low affinity for the autoantigen, as recently proposed by Cho et al. [45]. In their study, single-cell repertoire analysis of circulating Dsg3-specific memory B cells from pemphigus vulgaris patients revealed that these cells were mostly class-switched and affinity-matured. Interestingly, when monoclonal Dsg3-specific autoantibodies derived from patients were germline-reverted, they could no longer bind Dsg3, implying that these cells have acquired pathogenic autoreactive properties through SHM. Similarly, another study found that germline-reverted ACPAs lost or reduced reactivity to citrullinated proteins, suggesting a similar mechanism in RA [46]. The observation that germinal center-derived circulating autoreactive class-switched memory B cells, which are most likely affinity-matured cells, are enriched in GPA suggests also a similar mechanism in this disease [43].

A different mechanism for the development of autoreactive B cells has been proposed for SLE. Analysis of the circulating nuclear antigen-specific B cell frequencies, within the different B cell subsets, demonstrated that the nuclear antigen-specific B cells in SLE patients are eliminated to the same extent as the nuclear antigen-specific B cells in healthy individuals, suggesting properly functioning central and peripheral tolerance checkpoints in SLE [47]. However, anergy induction of circulating naive nuclear antigen-specific B cells in SLE patients appeared to be defective since the frequency of circulating anergic cells (IgM^low^) within the naive nuclear antigen-specific B cell compartment was found to be significantly lower compared to healthy individuals [47]. Moreover, de la Varga-Martiınez and colleagues observed a higher frequency of CD69^+^ activated B cells within the circulating nuclear antigen-specific B cell compartment of SLE patients compared to the total B cell compartment of the same patients [48]. Intriguingly, the activated nuclear antigen-specific B cells were mostly of the naive subset. These studies indicate that a fraction of naive nuclear antigen-specific B cells in SLE possibly escape early tolerance checkpoints due to failure of anergy induction and can be activated. This suggests that activated naive autoreactive B cells may be the precursor cells of autoreactive B cells in SLE. However, tolerance could also be breached in germinal centers, and autoreactive B cells can be generated from non-autoreactive B cells/non-pathogenic autoreactive B cells with a low affinity for autoantigens, since germline-reverted autoantibodies of SLE patients were demonstrated to be unreactive to SLE-related autoantigens [49,50].

Together, a breach of tolerance checkpoints in germinal centers appears to be a common mechanism that permits autoreactive B cells development across different autoimmune diseases. Moreover, current evidence regarding the precursors of autoreactive B cells, in diseases with high-affinity autoantibodies, points to non-autoreactive B cells/non-pathogenic autoreactive B cells with low affinity for the autoantigen that are further selected for high affinity in germinal centers after SHM. In SLE, early checkpoints may be additionally breached, and autoreactive B cells may also derive from activated naive autoreactive B cells, in which anergy induction is impaired.

### 3.2. Activation of Autoreactive B Cells

T cell-dependent activation of B cells requires the availability of antigens that can bind to the BCR and are subsequently internalized, processed, and presented by MHC-II molecules on the cell surface to activate antigen-specific T_H_ cells. Additionally, stimulation of innate immune receptors, such as TLRs and CRs, can help to promote the activation of B cells. Data on the activation of autoreactive B cells, and whether this deviates from normal B cell activation, are scarce since most studies have not specifically investigated activation of the autoreactive B cells themselves. Therefore, we summarize the activation mechanisms mainly from studies that have investigated these mechanisms indirectly.

B cells encounter extracellular antigens circulating through the secondary lymphoid organs or presented by specialized APCs that present antigens in their native state [51]. Autoantigens can be present at various intra- and extracellular locations depending on the disease type and target autoantigen(s) (Table 1). RA is an example in which autoantigens are persistently available since most citrullinated peptides, including citrullinated fibrinogen and type II collagen, are located extracellularly [52,53]. However, in many autoimmune diseases, the autoantigens are usually not located extracellularly and therefore not available to autoreactive B cells. Infection, injury, and inflammation are likely important factors for increasing the availability of such autoantigens. These factors cause cellular damage and the release of autoantigens. Importantly, disruptive cellular debris clearance, necrosis, and impaired degradation of neutrophil extracellular traps have been associated with increased susceptibility to autoimmune diseases due to an increased presence of autoantigens extracellularly [54]. In SLE in particular, it has been reported that apoptotic cell clearance is deficient, due to decreased phagocytosis, leading to secondary necrosis of apoptotic cells [55]. Subsequently, intracellular antigens are released. In addition, degradation of neutrophil extracellular traps is impaired in SLE causing intracellular antigens to be present longer extracellularly [55]. Moreover, intracellular autoantigens can be released by cells during inflammatory processes as part of their normal function. As an example, the autoantigen in GPA, proteinase 3 (PR3), is a protease involved in the degradation of extracellular matrix proteins, which is released from neutrophils upon activation during inflammation, thus exposing PR3-specific B cells to their autoantigen [56,57,58].

Since T cell-dependent activation of B cells requires T cell help for complete activation, T cells with the same antigen specificity as the autoreactive B cells should be present in autoimmune diseases to support autoreactive B cell activation. Autoantigen-specific T cells have been detected in several autoimmune diseases including RA, SLE, GPA, and pemphigus vulgaris [59,60,61,62]. In RA patients for example, T cells reactive against various RA-associated citrullinated peptides have been detected in the peripheral blood and lymphoid tissues [59]. For the detection, tetramers were used consisting of citrullinated peptides bound to HLA-DRB1*04:01, which is encoded by an HLA class II RA risk allele associated with binding and presenting citrullinated peptides. The binding of autoantigen-MHC-II to T cells suggests that T cells are present that recognize autoantigens displayed by MHC-II, thus having the same antigen specificity as autoreactive B cells. These studies suggest that, in these autoimmune diseases, in addition to a breach in B cell tolerance, T cell tolerance has failed as well.

B cells express various pattern recognition receptors including several TLRs, which act as co-receptors for the BCR. Among the TLRs expressed by B cells, activation of TLR7 and TLR9 has been associated with autoimmunity [63]. TLR7 and TLR9 are located in the endosomal compartment of cells and are activated upon binding of ssRNA and DNA with unmethylated CpGs, respectively [64]. Synergistic activation of the BCR and TLR7 or TLR9 with nuclear antigens has been shown to activate autoreactive rheumatoid factor (RF)-specific murine B cells [65,66,67]. Intriguingly, TLR7/TLR9 activation and BAFF overexpression have been associated with the activation of autoreactive B cells in SLE-like mouse models independent of T cells [68,69]. Since autoantigens in SLE are mostly ligands for both the BCR and TLRs, it can be hypothesized that stimulation by such antigens alone could be sufficient for the activation of autoreactive B cells independent of T cell signals [53]. These data implicate that extracellular nuclear antigens, when available, may contribute to autoreactive B cell activation by simultaneously activating TLRs.

Taken together, T cell-dependent activation of autoreactive B cells can be initiated, depending on the autoimmune disease, by autoantigens that are persistently expressed extracellularly. However, in most diseases, the autoantigens are not located extracellularly and factors such as infection and defective apoptotic cell clearance are responsible for autoantigens to be released and become accessible for autoreactive B cells. The observation that autoantigen-specific T cells are found in many autoimmune diseases suggests that in these diseases autoreactive B cell-autoantigen-specific T cell interaction can take place for the activation of autoreactive B cells. Furthermore, activation of TLRs expressed by B cells may aid in autoreactive B cell activation and possibly can activate these cells in a T cell-independent fashion, for example in SLE.

### 3.3. Escape of Tolerance

Self-tolerance is compromised in autoimmunity, but how autoreactive B cells escape self-tolerance checkpoints is largely unknown. Recent studies have shed some light on this issue.

Genetic predisposition and environmental triggers are known to play a central role in the breakdown of tolerance, leading to the escape of autoreactive B cells from tolerance mechanisms. Various genetic variants of HLA genes, B/T cell signaling genes, stimulatory/inhibitory signaling pathways, and cytokine/cytokine receptor genes have been identified as risk factors associated with loss of self-tolerance and development of autoimmunity [70]. To illustrate the role of genetic susceptibility, Joshua et al. analyzed the effect of carriage of the genetic variant of the PTPN22 gene (R620W) on the frequency of circulating citrullinated fibrinogen peptide-specific B cells in RA patients [71]. PTPN22 is a protein tyrosine phosphatase involved in BCR signaling. The R620W variant of the PTPN22 gene has been identified as a strong risk factor for several autoimmune diseases, possibly by affecting BCR signaling and selection of autoreactive B cells during development [72]. The authors reported a trend towards increased frequencies of circulating citrullinated fibrinogen peptide-specific B cells in RA patients carrying the R620W variant compared to non-R620W-carrying RA patients. Thereby, they suggested a link between genetic risk factors and breach of B cell self-tolerance.

Aberrant BCR signaling, in particular enhanced Bruton’s tyrosine kinase (BTK) activity, is possibly implicated in the breach of self-tolerance in autoimmunity. Components of the BCR signaling pathway control the activation threshold of B cells, therefore they play a crucial role in maintaining self-tolerance. BTK is a tyrosine kinase in the BCR signaling pathway that plays a central role in transducing BCR signals into downstream signals which result in B cell activation and survival [73]. It has been shown that autoimmunity is induced in transgenic mice overexpressing human BTK in B cells and that inhibition of BTK is effective in reducing or preventing autoimmunity in various murine autoimmune models [74]. These mouse studies suggest that BTK overexpression can breakdown B cell tolerance [74,75]. Interestingly, compared to healthy individuals, elevated BTK levels were observed in circulating B cell subsets, including naive B cells, of patients with ACPA-positive RA, Sjögren’s syndrome, and active GPA [76,77]. Moreover, it has been demonstrated in active GPA patients that the B cell subsets with elevated BTK levels (transitional and naive B cells) are hyperresponsive to BCR stimulation [77]. This could indicate that elevated BTK levels may increase BCR sensitivity, thus lowering the activation threshold of B cells. Furthermore, BTK is associated with contributing to disease pathogenesis since BTK levels of B cells were correlated with serum RF-antibody levels, and the degree of salivary gland T cell infiltration in Sjögren’s syndrome and ACPA positivity in RA patients [76]. Together, these studies demonstrate that elevated BTK levels are a characteristic of B cells across different autoimmune diseases. Elevated BTK levels could lower the activation threshold of B cells, suggesting that B cells with elevated BTK levels possess the capacity to breach tolerance.

A possible mechanism by which autoreactive B cells can escape from tolerance checkpoints in germinal centers is by the introduction of new *N*-linked glycans in the variable (Fab) region of BCRs, which potentially influence autoreactive B cell selection. *N*-linked glycosylation is a process in which *N*-linked glycans are attached to asparagine (*N*) sites of proteins [78]. SHM can introduce new *N*-linked glycosylation sites into the Fab region which are not present in germline-encoded antibodies, causing new glycosylation patterns [79,80,81]. It has been estimated that 15–25% of serum IgG in healthy individuals is Fab glycosylated [78,82,83]. Intriguingly, a hyperglycosylated Fab region pattern has been observed in several autoimmune diseases. In RA, it has been shown that more than 90% of ACPA-IgG in serum carry *N*-linked glycans in their Fab region, while this is only around 17% of ACPA-depleted IgG from the same patients [82]. In Sjögren’s syndrome, it was demonstrated that 24% of serum IgG carry *N*-linked glycans in the Fab region [84]. The exact mechanisms by which Fab glycosylation can alter selection during germinal center reactions, and thereby could aid in escaping tolerance checkpoints, are still elusive. However, it has been proposed that Fab glycosylation introduced by SHM in autoreactive B cells might change BCR affinity for antigens, which can give these cells a selective advantage when antigen binding affinity is increased [79,85]. Moreover, the interaction of Fab glycans with lectins derived from the microenvironment may cause crosslinking of BCRs. Crosslinking of BCRs can provide a survival signal, thus may give a survival advantage to autoreactive B cells [79,85].

It cannot be excluded that autoreactive B cells display distinct gene expression profiles which make them more prone to escape self-tolerance. Indeed, studies have demonstrated that in several autoimmune diseases the autoreactive B cells show a distinct gene expression pattern compared to non-autoreactive B cells. In one study, the expression profile of Bcl-6/Bcl-xL-immortalized CCP2-specific memory B cells, generated from peripheral blood or synovial fluid of RA patients, was assessed by RNA-sequencing and/or flow cytometry [86]. Bcl-6/Bcl-xL-immortalized CCP2-specific memory B cells showed higher expression of the costimulatory receptors CD40 and C5aR1 compared to Bcl-6/Bcl-xL-immortalized non-CCP2-specific memory B cells generated from the same patients. In another study, higher expression of IL-15Rα and amphiregulin (AREG; an epidermal growth factor receptor ligand) was observed in circulating CCP1-specific B cells of RA patients compared to the circulating non-CCP1-specific B cells from the same patients [87]. Since costimulatory signals, cytokine receptors, and growth factors are crucial factors in B cell development and self-tolerance, it is tempting to speculate that their increased expression by autoreactive B cells contributes to the escape of tolerance induction. Theoretically, increased expression of costimulatory molecules could enhance B-T_H_/T_FH_ cell interactions, providing a survival advantage and causing increased activation, proliferation, and differentiation of autoreactive B cells, in comparison with non-autoreactive B cells. Moreover, the upregulation of cytokine receptors and growth factors could have similar effects. Collectively, these factors may promote the breach of self-tolerance by autoreactive B cells. While this an interesting concept, research has yet to be conducted to causally link distinct genetic traits of autoreactive B cells and breach of self-tolerance.

Overall, various mechanisms have been proposed by which autoreactive B cells may breach self-tolerance checkpoints. These mechanisms include a break of self-tolerance influenced by genetic and environmental factors, enhanced BTK activity, altered glycosylation patterns of autoreactive BCRs, and possibly aberrant expression of stimulatory receptors by autoreactive B cells.

### 3.4. Effector Functions of Autoreactive B Cells

Autoreactive B cells that have escaped self-tolerance mechanisms may drive the inflammatory manifestations through the exertion of various effector functions. Rituximab targets CD20, which is expressed by B cells from the pre-B cell stage through the pre-plasma stage, thus excluding the cells that produce antibodies, i.e., plasmablasts and plasma cells [88]. Clinical outcomes following rituximab provide insights into the functional characteristics of autoreactive B cells.

Autoantibodies have an important effector function in some diseases. For example, in pemphigus, anti-Dsg1 and anti-Dsg3 antibodies bind to keratinocyte adhesion proteins desmoglein-1 and -3, respectively [11]. Consequently, adhesion is lost between keratinocytes, leading to blister formation in the skin or oral mucosa [11]. In SLE, ANAs bind to nuclear antigens and subsequently form antigen-antibody complexes [9]. These immune complexes deposit in various tissues, which cause inflammation and tissue damage [9]. Given that rituximab targets CD20, the contribution of short-lived plasma cells and long-lived plasma cells to autoantibody production can be identified. It has been demonstrated that rituximab is effective in diseases such as RA, GPA, and pemphigus and can lead to a reduction of autoantibody levels [89,90,91,92,93,94,95]. These observations indicate that short-lived plasma cells contribute to the production of pathogenic autoantibodies in these diseases. In pemphigus in particular, the high efficacy of rituximab, the dramatic reduction of anti-Dsg1/anti-Dsg3 serum levels following rituximab, and the correlation of these autoantibodies with clinical response indicate that short-lived plasma cells are the main source of autoantibodies in this disease [93,94,95]. In RA and GPA however, autoantibodies levels are not always drastically reduced following rituximab, implying that long-lived plasma cells also contribute to autoantibody production [89,92]. In contrast, long-lived plasma cells may be the major source of autoantibodies in diseases in which the efficacy of rituximab is low. Rituximab has low efficacy in Sjögren’s syndrome, an autoimmune disease characterized by the presence of anti-SSA(Ro) and anti-SSB(La) autoantibodies [96,97]. Various studies have demonstrated that rituximab does not impact anti-SSA(Ro)/anti-SSB(La) levels in patients supporting that long-lived plasma cells are likely the predominant autoantibody producers [98,99,100].

B cells may promote inflammatory processes through antibody-independent functions [5,6]. As an example, cytokine expression by autoreactive B cells has been linked to disease pathogenesis in pemphigus vulgaris. In pemphigus vulgaris patients, the frequency of IL-1β-expressing circulating Dsg1/Dsg3-specific B cells was significantly higher at baseline compared to healthy individuals and decreased to healthy control levels when in remission following rituximab treatment. Conversely, no change in the frequencies of IL-1β-expressing circulating non-Dsg1/Dsg3-specific B cells was observed in the same patients, suggesting that autoreactive B cells possibly mediate autoimmune processes by the secretion of specific cytokines [101]. Furthermore, as another example, crosstalk between autoreactive B cells and CD8^+^ T cells may promote pathogenic processes as well. While not specifically focused on the autoreactive B cells themselves, it has been demonstrated in GPA patients that rituximab lowered the frequency of circulating CD8^+^ T_EMRA_ cells, whereas the frequencies of different CD4^+^ T cell subsets and T_reg_ cells were unaffected. Importantly, rituximab reduced cytokine and chemokine production of CD8^+^ T cells and coculturing of B cells from untreated active GPA patients with CD8^+^ T cells from the same patients enhanced CD8^+^ T cell proinflammatory cytokine production, suggesting a pathogenic crosstalk between these cells [102].

Collectively, autoreactive B cells can drive autoimmune disease by pathogenic autoantibodies of which the contribution of antibodies produced by short-lived versus long-lived plasma cells vary between diseases. Additionally, autoreactive B cells may influence disease processes by antibody-independent mechanisms, such as by the production of specific cytokines and crosstalk with T cells.

## 4. Implications for Therapy and Perspectives

Currently, clinical remission is often successfully induced in most patients through broad suppression of the immune system by corticosteroids, immunosuppressants, and biologics. Despite remission induction, many patients experience relapses upon cessation of therapy. Moreover, chronic treatment of patients with broad immune-suppressing agents may lead to adverse effects such as increased susceptibility to infections and diminished effectiveness of vaccines. Therefore, there is a need for novel therapeutic strategies that more specifically target autoreactive B cells and prevent the reconstitution of these cells.

Intervening with the mechanisms involved in the breach of self-tolerance is crucial for preventing autoreactive B cell development. The fact that tolerance checkpoints in germinal centers are breached among various autoimmune diseases suggests that restoring these checkpoints could hold promise for preventing autoreactive B cell development. Since tolerance of B cells is regulated by BCR signaling thresholds, targeting modulators of B cell signaling may restore tolerance. Some important inhibitory checkpoints of B cells that could be targeted are for example programmed cell death 1 (PD1), low-affinity immunoglobulin-γ Fc region receptor IIb (FcγRIIb), and CD22 [103]. Examples of important stimulatory checkpoints are CD40, TLRs, and B cell-activating factor receptor (BAFFR) [103]. Furthermore, the finding that BTK levels of B cells are elevated in various autoimmune diseases, and BTK overexpressing is associated with breaching tolerance and development of autoimmunity, makes BTK a potential therapeutic target. Inhibition of BTK may enhance the activation threshold of autoreactive B cells, thereby preventing the breach of tolerance. Clinical trials are currently underway to investigate the effect of BTK inhibition in autoimmune diseases including RA, SLE, pemphigus, Sjögren’s syndrome, and multiple sclerosis (www.clinicaltrials.gov, accessed on 12 May 2021). These trials will soon provide information regarding the efficacy of BTK inhibition. Moreover, Fab glycosylation may also be an interesting therapeutic target, although more research is needed to decipher the exact role of Fab glycosylation in autoimmunity.

The observation that autoreactive B cells have a distinct gene expression profile compared to autologous non-autoreactive B cells is interesting and may provide clues to new targets for therapy, although it is unlikely that such targets will be specific for autoreactive B cells. Recently, novel cell-based therapies have been developed for the treatment of autoimmune diseases that aim to specifically eliminate autoreactive B cells [104,105]. Chimeric autoantibody receptor T (CAAR-T) cells are T cells that are genetically engineered to express a chimeric receptor that consists of an extracellular antigen domain fused to T cell cytoplasmic signaling domains. CAAR-T cells are activated upon binding of B cells to the extracellular antigen domain, resulting in cytotoxic T cell-mediated elimination of B cells. Ellebrecht and colleagues have shown that Dsg3-CAAR-T cells could be used to specifically target Dsg3-specific B cells for the treatment of pemphigus vulgaris. Dsg3-CAAR-T cells were constructed of which the extracellular domain of the chimeric receptor consisted of Dsg3 and the intracellular signaling domains of CD137-CD3ξ. In vitro, they showed that Dsg3-CAAR-T cells had specific cytotoxic activity against Dsg3-specific hybridomas. In vivo, Dsg3-CAAR-T cells also had specific cytotoxic activity against Dsg3-specific hybridomas in NOD-scid-gamma mice, which were injected with a mix of Dsg3-specific hybridomas. In addition, Dsg3-CAAR-T cell treatment of mice lowered Dsg3-specific IgG autoantibody levels and prevented the formation of mucosal blisters [106]. In a subsequent pre-clinical study, they demonstrated that Dsg3-CAAR-T cells could also specifically eliminate circulating Dsg3-specific B cells from patients ex vivo [107]. Zhang and colleagues used a different approach to specifically deplete autoreactive B cells. In this approach, a combination of chimeric antigen receptor T (CAR-T) cells and fluorescently labeled autoantigens was used. Anti-fluorescein isothiocyanate (FITC) CAR-T cells, combined with various FITC-labeled RA-related citrullinated protein epitopes, were found to specifically eliminate hybridoma cells in vitro and autoreactive B cells from RA patients ex vivo [108].

Together, possible therapeutic targets for depleting autoreactive B cells and/or preventing the reconstitution of these cells include stimulatory and inhibitory checkpoints of B cells, BTK, Fab glycosylation, and targets based on the unique gene expression profile of autoreactive B cells. Ultimately, CAAR-T cell- and CAR-T cell-based strategies may be the most attractive approach for eliminating autoreactive B cells without targeting non-pathogenic cells (Figure 3).

## 5. Conclusions

Overall, multiple mechanisms are likely involved in the development and activation of autoreactive B cells in B cell-mediated autoimmune diseases. Some mechanisms overlap between different diseases, suggesting also overlap in potential targets for therapy. Eventually, deciphering these mechanisms may further lead to the identification of specific targets as a basis for novel therapeutic strategies for autoimmune diseases.

## Figures and Tables

**Figure 1 cells-10-01190-f001:**
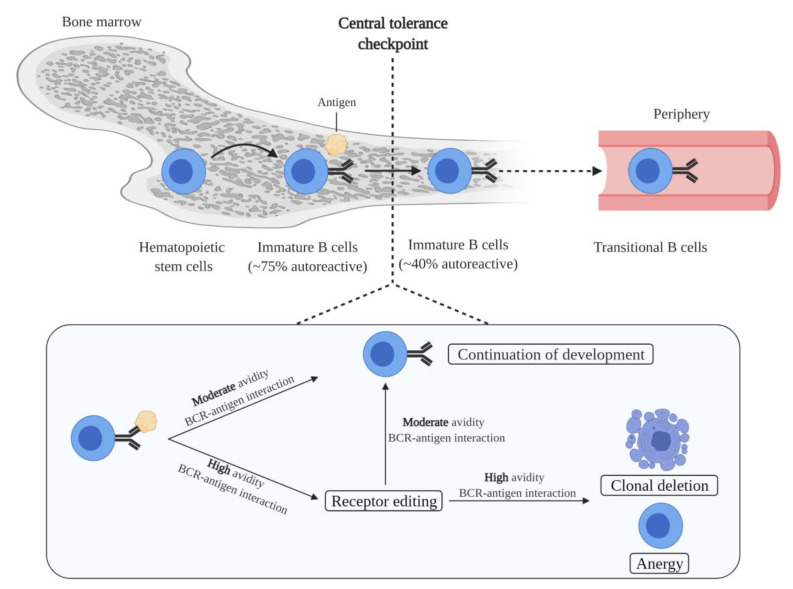
Elimination of autoreactive B cells by central tolerance mechanisms. The development of immature B cells from hematopoietic stem cells is accompanied by the generation of autoreactivity as the result of random variable (V), diversity (D), and joining (J) recombination. In the bone marrow, a significant proportion of autoreactive immature B cells is reduced at the central tolerance checkpoint. Central tolerance mechanisms include clonal deletion, anergy, and receptor editing. Central tolerance mechanisms are induced dependent on the binding strength between the BCR of immature B cells and self-antigens present in the bone marrow. Immature B cells that have escaped central tolerance mechanisms migrate into the periphery as transitional B cells. Figure was created with BioRender.

**Figure 2 cells-10-01190-f002:**
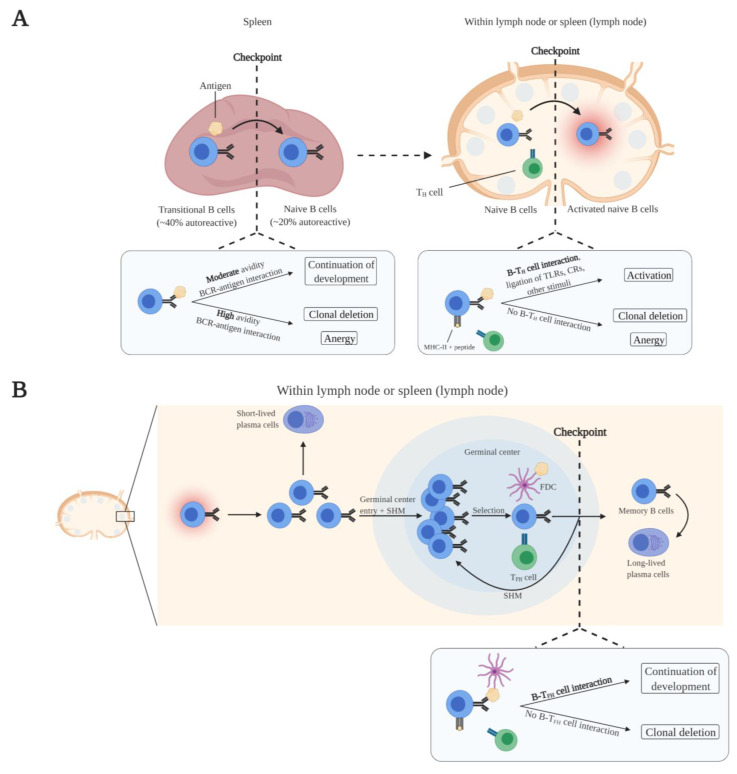
Elimination of autoreactive B cells by peripheral tolerance mechanisms at various checkpoints. (**A**) Maturation of transitional B cells takes place in the spleen. Transitional B cells that strongly bind self-antigens present in the spleen undergo clonal deletion or anergy, which reduces the frequency of autoreactive B cells. The transitional B cells that moderately bind self-antigens mature into naive B cells. Naive B cells predominantly encounter antigens within lymph nodes and the spleen. Activation of naive B cells is dependent on the binding of antigens and interaction with CD4^+^ T helper (T_H_) cells with the same antigen specificity in which B cells receive costimulatory signals. Naive B cells that do not have interaction with T_H_ cells undergo clonal deletion or anergy, which further reduces the frequency of autoreactive B cells. (**B**) Activated B cells enter germinal centers within lymph nodes and the spleen, undergo somatic hypermution (SHM) and isotype switching, and ultimately mature into memory B cells and long-lived plasma cells. These maturation processes are dependent on costimulatory signals from CD4^+^ follicular T helper (T_FH_) cells with the same antigen specificity. Clonal deletion is induced in B cells that do not receive costimulatory signals from T_FH_ cells, resulting in the removal of autoreactive B cells from the B cell pool. Figure was created with BioRender.

**Figure 3 cells-10-01190-f003:**
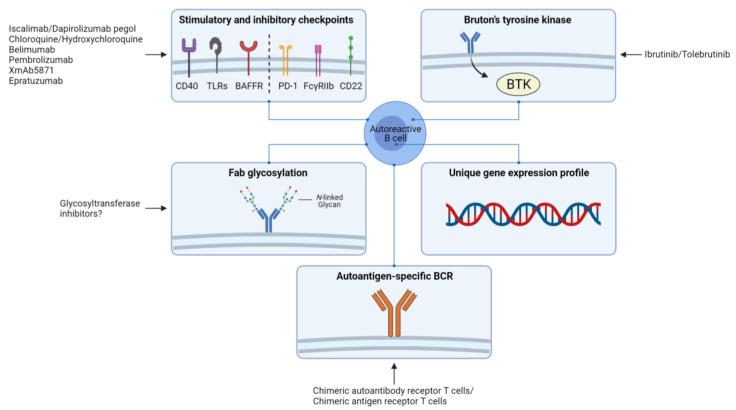
Overview of potential therapeutic approaches for targeting autoreactive B cells. Stimulatory and inhibitory checkpoints of B cells such as CD40, Toll-like receptors (TLRs), B cell-activating factor receptor (BAFFR), programmed cell death 1 (PD1), low-affinity immunoglobulin-γ Fc region receptor IIb (FcγRIIb), and CD22; Bruton’s tyrosine kinase (BTK); glycosylation patterns of the variable (Fab) region of the B cell receptor (BCR); targets based on the unique gene expression profiles of autoreactive B cells compared to autologous non-autoreactive B cells; and the autoantigen-specific BCR of autoreactive B cells. Figure was created with BioRender.

**Table 1 cells-10-01190-t001:** Summary of various classical B cell-mediated autoimmune diseases.

Disease	Major Affected Tissue(s)	Predominant Autoantigen(s)	Location of Autoantigen(s)	Predominant Autoantibodies	B Cell-Depleting Therapy Effective?	References
Rheumatoid arthritis (RA)	Joints	Cyclic citrullinated peptides/proteins (CCP), IgG	Intra- and extracellular	Anti-citrullinated protein antibodies (ACPAs), rheumatoid factor (RF)	Yes	[7,8]
Systemic lupus erythematosus (SLE)	Multiple	Double-stranded DNA, Smith, SSA(Ro), SSB(La)	Nucleus	Anti-nuclear (ANA), anti-double-stranded DNA, anti-Smith, anti-SSA(Ro), anti-SSB(La)	Inconclusive	[4,9]
Granulomatosis with polyangiitis (GPA)	Airways, kidneys	Proteinase-3 (PR3)	Cytoplasm	PR3-anti-neutrophil cytoplasmic antibody (PR3-ANCA)	Yes	[10]
Microscopic polyangiitis	Kidneys, skin	Myeloperoxidase (MPO)	Cytoplasm	MPO-ANCA	Yes	[10]
Pemphigus vulgaris	Oral mucosa and/or skin	Desmoglein-3 (Dsg3), Desmoglein-1 (Dsg1)	Cell surface	Anti-Dsg3, Anti-Dsg1	Yes	[11]
Pemphigus foliaceus	Skin	Dsg1	Cell surface	Anti-Dsg1	Yes	[11]
Bullous pemphigoid	Skin	BP180 BP230	Cell surface Intracellular	Anti-BP180 Anti-BP230	Yes	[12,13]
Sjögren’s syndrome	Salivary glands, lacrimal glands	SSA(Ro), SSB(La)	Nucleus	Anti-SSA(Ro), anti-SSB(La)	Inconclusive	[14]
Myasthenia gravis	Muscles	Acetylcholine receptor (AChR), muscle-specific kinase (MuSK)	Cell surface	Anti-AChR, anti-MuSK	Yes	[15]
Immune thrombocytopenia	Platelets	Platelet glycoprotein (GP) IIb/IIIa, GPIb-IX-V	Surface of platelets	Anti-Ib/IIIa, anti-GPIb–IX-V	Yes	[16,17]
Graves’ disease	Thyroid gland	Thyroid-stimulating hormone receptor (TSHR)	Cell surface	Anti-TSHR	Inconclusive	[18]
Anti–glomerular basement membrane disease	Kidneys, lungs	Type IV collagen	Extracellular	Anti–glomerular basement membrane antibody	Inconclusive	[19]
Multiple sclerosis	Central nervous system	Unknown	-	Unknown	Yes	[20]
Chronic inflammatory demyelinating polyneuropathy	Peripheral nervous system	Contactin-1, neurofascin-155/140/186	Cell surface	Anti-contactin-1, anti-neurofascin-155/140/186	Possibly	[21,22]
Guillain–Barré syndrome	Peripheral nervous system	GD1a, GM1b, GM1, GD1b, GT1a GalNac-GD1a, GQ1b	Cell surface	Anti-ganglioside	Unknown	[23,24]

## Data Availability

Not applicable.

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
