# Peer review of "B Cell Activation and Escape of Tolerance Checkpoints: Recent Insights from Studying Autoreactive B Cells"

_cells, 2021, doi:10.3390/cells10051190_

Round 1
Reviewer 1 Report
Well-written review, up-to-date, and important to publish. I suggest to add additional autoimmune diseases to the list in table 1, e.g., pemphigoid diseases of the skin (including bullous pemphigoid).
Author Response
We would like to thank the reviewer for evaluating our manuscript. We are pleased with his/her positive comments and appreciate his/her suggestion. We have added three B cell-mediated autoimmune diseases to the list in table 1: Bullous pemphigoid, Immune thrombocytopenia, and Graves’ disease.
Reviewer 2 Report
Bonasia CG et al submitted a review entitled « B cell activation and escape of tolerance checkpoints : recent insights from studying autoreactive B cells ». Most of the authors of this review are recognized in the field of auto-immune disease. This review is well written and well documented and provides a very interesting overview of the literature in the field.
Minor points:
Figure 2: Labels are lacking. The order of the figure has been inversed. The lymph node is not very visible
Figure 3: Figure legend is lacking. Therapeutic strategies should be added.
Author Response
We would like to thank the reviewer for reviewing our manuscript and we appreciate his/her positive comments. We have addressed his/her comments regarding the figures. Various adjustments have been made to the figures:
Figure 2:
- The order of the figure has been reversed. First, the overview of tolerance checkpoints in the spleen and the lymph node is shown, and second the overview of tolerance checkpoints in the germinal center.
- Label ‘A’ has been added to the overview of tolerance checkpoints in the spleen and the lymph node. Label ‘B’ is added to the overview of tolerance checkpoints in the germinal center.
- The opacity of the lymph nodes in 2A and 2B has been enhanced to increase their visibility.
Figure 3:
- A figure legend has been added.
-Therapeutic strategies have been added for various potential therapeutic approaches for targeting autoreactive B cells (‘Stimulatory and inhibitory checkpoints’, ‘Bruton’s tyrosine kinase’, ‘Fab glycosylation’, and ‘autoantigen-specific BCR’).